# Generative Adversarial Network-Based Anomaly Detection and Forecasting with Unlabeled Data for 5G Vertical Applications

**Qing Zhang [1,2], Bin Chen [1], Taoye Zhang [1], Kang Cao [1], Yuming Ding [1], Tianhang Gao [1] and Zhongyuan Zhao [2,*]**

[1] Intelligent Network Innovation Center, China Unicom, Beijing 100048, China; zhangq49@chinaunicom.cn (Q.Z.); chenbin12@chinaunicom.cn (B.C.); zhangtaoye@chinaunicom.cn (T.Z.); caok6@chinaunicom.cn (K.C.); dingym19@chinaunicom.cn (Y.D.); gaoth11@chinaunicom.cn (T.G.)

[2] The State Key Laboratory of Networking and Switching Technology, School of Information and Communications Engineering, Beijing University of Posts and Telecommunications, Beijing 100876, China

[*] Correspondence: zyzhao@bupt.edu.cn

**Abstract:** With the development of 5G vertical applications, a huge amount of unlabeled network data can be collected, which can be employed for evaluating the user experience and network operation status, such as the identifications and predictions of network anomalies. However, it is challenging to achieve highly accurate evaluation results using the conventional statistical methods due to the limitations of data quality. In this paper, generative adversarial network (GAN)-based anomaly detection and forecasting are studied for 5G vertical applications, which can provide considerable detection and prediction results with unlabeled network data samples. First, the paradigm and deployment of the deep-learning-based anomaly detection and forecasting scheme are designed. Second, the network structure and the training strategy are introduced to fully explore the potential of the GAN model. Finally, the experimental results of our proposed GAN model are provided based on the practical unlabeled network operation data in various 5G vertical scenarios, which show that our proposed scheme can achieve significant performance gains for network anomaly detection and forecasting.

**Keywords:** anomaly detection and forecasting; 5G vertical application; unlabeled data; network quality; neural networks

## 1. Introduction

The transition of the industrial network infrastructure in the era of Industry 4.0 requires that diverse communication requirements are fulfilled, including those of smart cities, manufacturing technologies, and Industrial Internet of Things (IIoT) applications [1,2]. With the advent of 5G networks, there is great potential to fulfill these requirements by offering high-speed connectivity, ultra-low latency, and massive capacity [3]. However, the heterogeneity of vertical industries introduces various application scenarios such as healthcare, smart grids, and intelligent transportation, each with diverse business characteristics and network demands. These differences pose challenges for the successful implementation and optimization of 5G networks in industrial environments [4,5]. As a result, optimizing 5G networks for industrial settings requires tailored approaches that consider the specific requirements of each vertical sector. Efficient resource allocation, traffic management, and quality of service (QoS) provisioning become critical factors that need to be addressed to maximize the benefits of 5G technology in industrial applications [6]. Developing optimization algorithms that can account for the interplay between the network infrastructure, industrial applications, and dynamic workloads is essential for achieving optimal network performance and meeting the unique demands of different industrial verticals. Future research efforts should focus on developing innovative solutions for network monitoring and optimization in industrial environments [7]. This includes exploring techniques for data labeling and traffic analysis in industrial networks, as well as designing adaptive

and dynamic optimization algorithms that can adapt to the changing needs of industrial environments.

The large-scale connectivity of IoT devices in 5G vertical applications introduces additional complexity to network monitoring. The sheer number of connected devices, each generating data and contributing to the overall network traffic, presents a challenge in effectively managing and analyzing the vast amount of information [8]. The diversity and differentiation of 5G services and terminals further complicate the monitoring process. Each vertical industry may have unique requirements and network demands, necessitating tailored monitoring approaches to ensure their optimal performance and meet specific needs [9]. Moreover, the dynamic nature of 5G networks and the real-time requirements of certain vertical industries pose further challenges [10]. For instance, industries such as intelligent transportation or healthcare demand real-time monitoring to enable timely decision making and ensure the safety and efficiency of operations. Meeting these high real-time requirements and providing actionable insights in a timely manner presents a significant technical challenge for network monitoring in 5G vertical applications. However, these challenges also present opportunities for advancements in network monitoring techniques. Innovative approaches such as machine learning (ML), artificial intelligence (AI), and data analytics can be leveraged to analyze unlabeled network data and effectively detect anomalies [11]. By harnessing the power of these technologies, operators can gain valuable insights into network performance, detect potential issues in advance, and take proactive measures to optimize network resources and ensure reliable and efficient operations.

In the realm of anomaly detection for time series network data, the identification of unusual behavior within a sequence of interconnected data is of the utmost importance. However, one of the major challenges faced in this field is the lack of anomaly labeling, making supervised learning approaches infeasible. To tackle this challenge, researchers have developed various unsupervised techniques in recent years. These techniques aim to detect anomalies without relying on labeled examples. Among the commonly utilized unsupervised methods, there are distance-based techniques such as K-nearest neighbors [12–16]. This approach measures the similarity between data points based on their distances, and anomalies are identified as points that significantly deviate from their neighboring data points. Clustering techniques, like K-means, group similar data points together and consider points that do not belong to any cluster as potential anomalies [17–19]. Classification techniques, such as one-class support vector machines (SVMs), create a model of normal behavior and classify instances that fall outside the learned boundaries as anomalies [20–22]. Probabilistic methods analyze the statistical properties of the data and identify instances that have low probability under the learned distribution as anomalies [23]. However, traditional unsupervised methods have faced limitations over time. One significant challenge is the exponential increase in the dimensionality and length of the acquired measurements in modern network systems. As the complexity of the data grows, traditional techniques struggle to effectively capture the intricate patterns and correlations present in the time series network data. Additionally, these methods may not perform optimally in capturing the temporal correlation across different time steps, which is crucial for accurate anomaly detection. The dynamic nature of network systems requires models to consider the sequential dependencies and changes over time, which traditional techniques may overlook [24].

As a result, there has been a shift towards employing deep learning-based unsupervised anomaly detection methods. These approaches leverage the power of deep neural networks to automatically learn representations and infer correlations between time series [25–28]. Recurrent neural networks (RNNs) [29–31], such as the long short-term memory (LSTM) network [32,33], have shown promise in capturing long-term dependencies and temporal correlations in time series data. These can effectively model sequential information and have been successfully applied to anomaly detection tasks. Additionally, generating adversarial networks (GANs) have been explored in this domain [34]. GANs can learn to generate synthetic data that closely resemble the normal behavior of the network

system, allowing for the detection of deviations from this learned distribution as anomalies. Despite the advantages of deep learning-based methods, they also face their own set of challenges. The training of RNNs can be computationally expensive and time-consuming due to the sequential nature of the network. GANs, on the other hand, can suffer from issues such as pattern collapse and non-convergence during training, which can impact their performance. Furthermore, deep learning-based methods often struggle with noisy data in multivariate time series, as noise can introduce false positives and reduce the accuracy of anomaly detection. Addressing these challenges is an active area of research.

To the best of our knowledge, ref. [25] proposed a novel model called MSCRED that addresses the challenges of anomaly detection, root cause identification, and anomaly severity interpretation in multivariate time series data. MSCRED specifically focuses on capturing the correlation between different variables in the time series, enabling more accurate anomaly detection. By jointly considering these tasks, MSCRED offers a comprehensive solution for analyzing complex network systems. Similarly, ref. [28] introduced OmniAnomaly, a multivariate time series recurrent neural network designed for stochastic anomaly detection. OmniAnomaly utilizes reconstruction probabilities to determine anomalies, providing a reliable and interpretable measure of anomaly severity. With its stochastic nature, OmniAnomaly is well suited for capturing uncertainties in time series data. However, one common limitation shared by these methods is the lack of consideration for the time cost associated with training. Training complex deep learning models can be computationally expensive and time-consuming, which may limit their practical applicability. To address this concern, ref. [35] introduced the unsupervised anomaly detection (USAD) method. USAD combines the power of autoencoders and generative adversarial networks (GANs) to achieve more stable and efficient training. By leveraging the strengths of both models, USAD improves the stability and speed of training for anomaly detection in time series data. While these methods offer valuable contributions to the field, none of them explicitly address the forecasting problem of future anomalies. Forecasting future anomalies is crucial for proactive risk management and preventive actions. This aspect is an important consideration for real-time monitoring and decision-making in dynamic network systems. Incorporating forecasting capabilities into anomaly detection models is an ongoing area of research and an important direction for further advancements in the field.

In this paper, we introduce an approach for detecting and forecasting anomalies in unlabeled network data recorded across various 5G vertical applications. As demonstrated in Figure 1, the data processing procedure of unlabeled network data is first executed, which is followed by two parallel modules, named anomaly detection and forecasting modules, respectively. In the anomaly detection module, a GAN-based detection model is designed to obtain the anomaly evaluation results. In the anomaly forecasting module, a prediction model is employed to analyze the relationship between the input network status parameters and the predicted data, and the final predicted anomaly results can be achieved by using the anomaly detection model. Network anomaly detection and forecasting are critical to guarantee the quality of services (QoS) of 5G vertical applications, especially for some developing regions, such as South America, where the telecommunication infrastructure construction is not adequate [36]. Our proposed scheme can significantly mitigate the impacts of network anomaly with the restrictions of capital and operational expenditure, which can accumulate the development of 5G vertical applications. The main contributions of this paper are as follows:

- First, to achieve anomaly detection in 5G vertical applications, the strengths of the autoencoder and the GANs are combined. The autoencoder efficiently distills the significant characteristics of the data, while GAN provides robustness by generating adversarial examples. This innovative combination greatly enhances both the accuracy and stability of our anomaly detection model, ensuring the precise identification and consistent performance across diverse data contexts.

- Second, in order to proactively prepare for anomalies, we employ an LSTM model to predict the 5G network quality data, followed by the use of unsupervised anomaly detection methods to identify anomalies in the predicted data. By leveraging the predictive power of the LSTM model, our approach seeks to anticipate anomalous behavior in advance, and subsequently mitigate any potential negative impacts on the network.
- Finally, the simulation results show that our method outperforms traditional anomaly detection algorithms. Moreover, our method demonstrates a remarkable accuracy in detecting and forecasting anomalies in unlabeled network data. Therefore, it has good performance in network quality monitoring for 5G vertical applications.

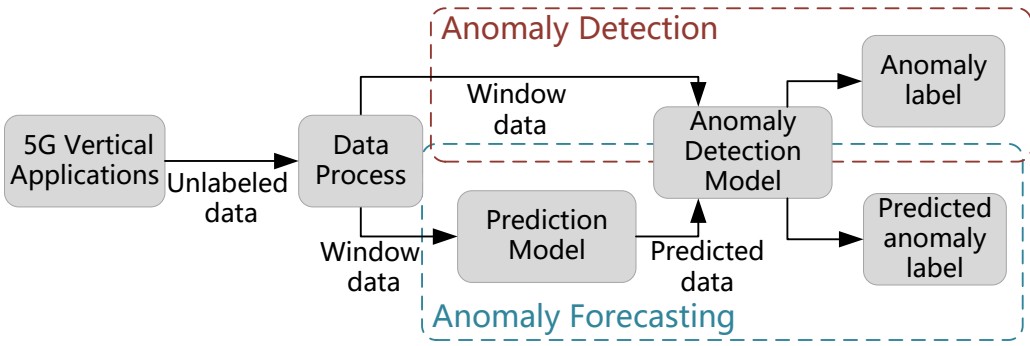

**Figure 1.** The illustration of the anomaly detection and forecasting problem.

The organization of this paper is as follows. In Section 2, we present the system model and formulate the associated problems related to network anomaly detection and forecasting. Section 3 provides a detailed exposition of the structure and algorithmic design underpinning our anomaly detection scheme. The structural design and algorithms of our anomaly forecasting scheme are the focus of Section 4. Simulation results and their corresponding discussions are offered in Section 5. Finally, Section 6 concludes the paper, summarizing key insights and findings.

## 2. System Model and Problem Formulation

We consider the network anomaly detection and forecasting problem in 5G vertical applications, which consists of three essential components:

- **Data process:** For the unlabeled network data, we express the set of time periods and that of data dimensions as $\mathcal{T} = \{1, \ldots, T\}$ and $\mathcal{M} = \{1, \ldots, M\}$, respectively. On this basis, the time series data of length $T$ can be expressed as $\mathcal{D} = \{\mathbf{d}_t\}_{t=1}^{T}$, where $\mathbf{d}_t \in \mathbb{R}^M$ denotes the network data at a specific time point $t$ with dimensions of $M$. To capture the relationship between the current time point and its previous ones, a processing procedure is necessary for the network data. This procedure transforms the data into a series of window data, denoted by $\mathcal{W} = \{\mathbf{w}_t\}_{t=1}^{T}$. Each window data at time point $t$ is defined as a sequence of $\mathbf{d}_t$ from $t - K + 1$ to $t$, expressed as $\mathbf{w}_t = \{\mathbf{d}_t\}_{t=t-K+1}^{t}$. It is worth noting that the length of each window data is $K$.
- **Anomaly detection:** Anomaly detection is applied to the processed window data $\mathcal{W}$ using an anomaly detection model. The underlying principle of anomaly detection involves training the model on known data samples and assessing the abnormality level of new data samples based on this learned representation. By establishing a threshold $\eta$, when the abnormality level of a new data sample surpasses the threshold at a given time point, it is classified as an anomaly. Hence, to identify the presence of anomalies in the new data samples, it is crucial to obtain a series of anomaly labels, which can be represented as $\mathcal{L} = \{l_t\}_{t=1}^{T}$, where $l_t$ indicates whether there is an anomaly in the sample at time point $t$.

- **Anomaly forecasting:** In contrast to anomaly detection, anomaly forecasting focuses on identifying anomalies in future time data. To accomplish this, the processed window data $\mathcal{W}$ is initially used as input for a prediction model, which generates the forecasts of the network data for the upcoming time points denoted by $\mathcal{P} = \{\mathbf{p}_t\}_{t=1}^{T}$. Subsequently, an anomaly detection model is applied to the predicted data, enabling the detection of abnormal patterns and providing a series of anomaly labels. Consequently, this approach facilitates accurate anomaly forecasting.

It is evident that the key aspect of both anomaly detection and anomaly forecasting problems revolves around identifying a suitable anomaly detection model, as anomaly forecasting essentially entails performing anomaly detection on the predicted data. With regard to the window data, assuming the historical data samples and new data samples are denoted by $\mathcal{W}$ and $\tilde{\mathcal{W}}$, respectively, the primary objective of the anomaly detection model is to determine the following anomaly scores, which enable the assessment of the abnormality level of the new data samples:

$$\mathcal{Y} = f(\mathcal{W}; \tilde{\mathcal{W}}) \tag{1}$$

After obtaining the aforementioned anomaly scores, it is possible to determine the abnormality of a specific time point by comparing the anomaly score, denoted by $\dagger_t$, with a threshold value $\eta$.

To achieve (1), traditional methods typically rely on sample classification and similar techniques. However, in the context of 5G vertical applications, network data often lack labeled information. To address this challenge, we propose a novel approach by combining autoencoder and generative adversarial network (GAN). Leveraging the reconstruction error from autoencoder and the adversarial framework of GAN, we successfully obtain anomaly scores.

For clarity, the main notations are summarized in Table 1.

**Table 1.** Summary of notations.

| Symbol | Description |
| --- | --- |
| $\mathcal{T}$ | The set of time periods |
| $\mathcal{M}$ | The set of data dimensions |
| $\mathcal{D}, \mathbf{d}_t$ | Time series data, $\mathbf{d}_t$ is the network data at a specific time point t |
| $\mathcal{W}, \mathbf{w}_t$ | Window data, $\mathbf{w}_t$ is the window data at a specific time point t |
| $\mathcal{L}, l_t$ | The series of anomaly labels, $l_t$ is the anomaly label at a specific time point t |
| $\mathcal{P}, \mathbf{p}_t$ | Forecasts of the network data, $\mathbf{p}_t$ is the forecast data at a specific time point t |

## 3. Network Anomaly Detection

In this section, we introduce the proposed anomaly detection model. Anomaly detection tasks currently face several challenges: (1) imbalanced class distribution with a large proportion of normal samples and a small number of anomalies, and (2) the lack of labeled information in network data. These challenges make it difficult to cover various types of anomalies that may arise in the future. To address these issues, the proposed anomaly detection model is a machine learning model that learns from the data itself, without the need for additional labels or prior knowledge about the data. In this section, we apply this approach for anomaly detection.

### 3.1. Data Acquisition and Process

To further enrich the characteristics of 5G vertical applications from multiple dimensions and to mine the features of different applications' business, traffic, and network

demands, it is first necessary to establish a contextual 5G vertical application feature library. This contextual feature library records the characteristics of network data in different application scenarios and their typical feature values, which allows for the consolidation of network data, the realization of unified network data management, and saves the data processing time, thereby enhancing the efficiency of subsequent analysis.

### 3.1.1. Feature Library Construction Framework

The need and significance of establishing a 5G vertical application network demand and traffic feature library based on the scenarios lies in the utilization of network data from dedicated network platforms. This seeks to establish a comprehensive feature evaluation method based on 5G private network business, network demands, and network traffic. Afterwards, based on the proposed feature evaluation method, this will facilitate the categorization of the network data features of 5G vertical applications in industry-specific scenarios. Ultimately, based on the 5G vertical application network demand and traffic feature library, recommendations for the deployment and planning of the 5G private network platform can be given. This not only guarantees network performance but also enhances the deployment efficiency and reduces the network planning costs.

In conclusion, the analysis of network demand and traffic features for contextual 5G vertical applications includes two requirements: (1) establish a multi-dimensional feature extraction and fusion method based on 5G private network business, network demand, and network traffic; (2) design a classification method based on multi-modal features of 5G private networks, and construct a 5G private network demand and traffic feature library model.

The industry-specific 5G private network traffic feature analysis model performs traffic feature analysis based on network-level data, and primarily consists of two parts. These are: a method for multi-dimensional feature extraction and fusion based on 5G private network business, network demands, and network traffic; and a method for constructing a 5G private network demand, business, and traffic feature library.

The multi-dimensional feature extraction and fusion scheme based on 5G private network business, network demand, and network traffic, processes the original data collected from the private network platform through a series of data-processing operations. This yields index data representing business features, network traffic features, and network demand. The 5G private network demand, business, and traffic feature library model, on the other hand, build a feature library and perform the related result analysis based on the dataset after the fusion of features, in accordance with the specific scenarios.

### 3.1.2. Feature Library Construction Methodology

Given the wealth of valid attributes within network data and the strong correlations amongst these features, optimal network traffic feature extraction cannot simply be realized based on coverage and information entropy. To satisfy the need for feature extraction that eliminates irrelevant and redundant features while completely reflecting network traffic characteristics, we build on business feature extraction based on information entropy screening. To facilitate more effective feature compression and selection, we chose to utilize the mutual information-based maximum relevance minimum redundancy (mRMR) coefficient algorithm.

The mRMR coefficient algorithm addresses a type of feature selection problem. The core idea of the algorithm is to find a group of features within the original feature data that exhibit maximum correlation with the final output results, but minimum redundancy amongst each other. The information quantity indicates how much information a feature contains, while the redundancy quantity characterizes the interrelation between features. The ultimate output of the algorithm, the mRMR coefficient, is the difference between these two quantities. Thus, during the execution of the algorithm, the effective features of overall network traffic are taken as inputs. In each iteration, the algorithm calculates the information quantity, redundancy quantity, and mRMR coefficient for each feature based

on mutual information metrics, ultimately yielding the mRMR coefficient for each network traffic feature.

The features in the network data are denoted by $\mathcal{F}_Y = \{f_1, f_2, \ldots, \}$. The computation of the mRMR coefficient via this algorithm primarily involves three steps. First, calculate the information quantity. The feature data with more information reflect the overall business characteristics better than the feature data with less information. Therefore, by comparing the information size of different features, different business features can be further filtered. Denoted by $I(f_i; f_n)$, the mutual information between the features $f_i$ and $f_n$, the information quantity can be expressed by

$$E(\mathcal{F}_Y, f_n) = \frac{1}{|\mathcal{F}_Y|} \sum_{f_i \in \mathcal{F}_Y} I(f_i; f_n). \tag{2}$$

Furthermore, less redundancy among features indicates less correlation between data. The redundancy quantity is given by

$$R(\mathcal{F}_Y) = \frac{1}{|\mathcal{F}_Y|^2} \sum_{f_i, f_j \in \mathcal{F}_Y} I(f_i; f_j). \tag{3}$$

The mRMR coefficient can be expressed as

$$r_n = E(\mathcal{F}_Y, f_n) - R(\mathcal{F}_Y). \tag{4}$$

As per the aforementioned algorithm, the group of features selected according to the mRMR coefficient maintain high correlation with the original data, thus ensuring that the output low-dimensional features can fully represent overall network traffic characteristics. Simultaneously, due to the smaller redundancy and less correlation between output features, the algorithm effectively eliminates irrelevant and redundant features, further realizing efficient feature compression.

### 3.2. The Structure of Network Anomaly Detection

Our proposed approach integrates the concept of an autoencoder and adversarial techniques, aiming to compute anomaly scores for network data at distinct time intervals via feature transformation. In detail, this approach incorporates two specific autoencoder models, referred to as the generating model $AE_1$ and the discriminating model $AE_2$. These models are involved in a mutual adversarial interaction designed to optimize their functions. The generator and discriminator are each composed of distinct decoder networks, namely $D_1$ and $D_2$, that operate in conjunction with a commonly shared encoder, $E$. For better comprehension, the architecture of the proposed network anomaly detection model is visually presented in Figure 2, with further detailed explanations provided subsequently.

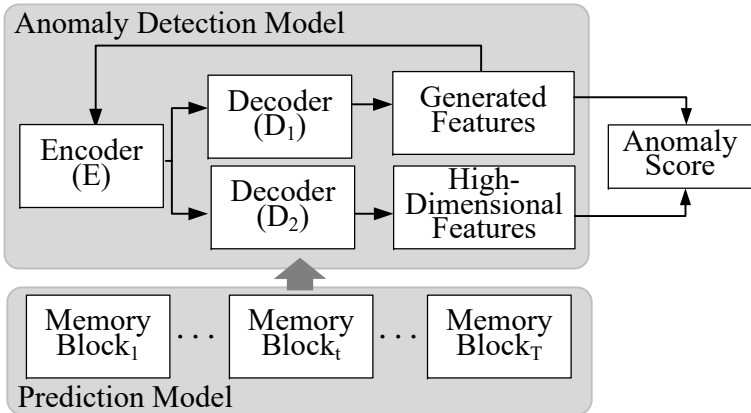

**Figure 2.** The architecture of the anomaly detection model and the prediction model.

### 3.2.1. Autoencoder-Based Feature Transformation

To start with, let us visualize the configuration composed of encoder $E$ and decoder $D_1$ as an individual autoencoder $AE_1$. When deploying an autoencoder for anomaly detection on unlabeled data, the input data serve as its own label. This prompts the neural network to decipher a mapping relationship, consequently creating a reconstructed output. An anomaly within the original input data can be suspected if the discrepancy between the reconstructed output and the original input surpasses a certain threshold. Two fundamental components encompass an autoencoder: the encoder and the decoder.

The encoder processes time window data $W$ as an input and primarily focuses on encoding the input data into a condensed latent representation $Z$. This process encourages the neural network to discern the most informative features. The output generated by the encoder can be characterized by a standard neural network function that is succeeded by the rectified linear unit (ReLU) activation function, as expressed by:

$$Z = ReLU(a_e \mathcal{W} + b_e). \tag{5}$$

where $ReLU(\cdot)$ stands for the ReLU activation function, while $a_e$ and $b_e$ represent the encoding weights and biases, respectively.

In order to reconstruct the latent representation back to its initial dimension, the decoder generates the reconstructed features, which can be mathematically formulated as:

$$\hat{\mathcal{W}} = ReLU(a_d Z + b_d), \tag{6}$$

where $\hat{\mathcal{W}}$ corresponds to the feature formed by the reconstruction, while $a_d$ and $b_d$ are the decoding weights and biases, respectively.

However, the decoder attempts to reconstruct the data as accurately as possible, thereby minimizing the reconstruction error $\left\|\mathcal{W} - \hat{\mathcal{W}}\right\|_2$. This endeavor could inadvertently overlook minor anomalies. Simply evaluating the possibility of anomalies by reconstructing the errors alone is not good, so the model also needs to be refined based on the idea of adversarial.

### 3.2.2. Enhancement via Adversarial-Based High-Dimensional Feature Transformation

Our methodology leverages the adversarial concept in conjunction with an autoencoder to bolster the detection sensitivity towards subtle anomalies. This section further elucidates how this integration operates and the key components involved in this process.

Specifically, $AE_2$ is composed of the encoder $E$ and the decoder $D_2$. This module functions as the discriminator, initiating an adversarial competition with $AE_1$. Analogous to $AE_1$, $AE_2$ also aims to reconstruct the input data, taking the generated features $\mathcal{W}$ delivered by $AE_1$ as input and providing high-dimensional features $\mathcal{W}'$ as output. However, a contrast to $AE_1$ is that $AE_2$ seeks to maximize the reconstruction error $\left\|\mathcal{W}' - \hat{\mathcal{W}}\right\|_2$.

In this configuration, the generator $AE_1$ strives to generate progressively realistic data, while the discriminator $AE_2$ improves its sensitivity and accuracy in distinguishing the original features from the generated features. Through this adversarial learning process, both the generator and discriminator mutually and iteratively refine their performances, fostering their capabilities via a competitive dynamic. This procedure ultimately augments the model's ability to discern subtle anomalies.

The integration of these components forms a composite model, exhibiting enhanced robustness. The model not only improves the anomaly detection accuracy of autoencoders, but also addresses potential issues such as training non-convergence and pattern collapse associated with the GANs.

In conclusion, the fusion of these processes and components forms an innovative approach, enhancing the model's overall robustness. The effectiveness of the autoencoder in anomaly detection is amplified, and challenges typically associated with GANs are effectively managed. This results in a more stable training process and improved performance in anomaly detection.

### 3.2.3. Acquisition of Anomaly Score

To obtain the anomaly score at any given time, we adopt a method based on the reconstruction error. This approach incorporates two key components, namely the reconstruction errors of $AE_1$ and $AE_2$, in the calculation of the anomaly score. Specifically, the score can be formulated as follows:

$$\mathcal{Y} = \alpha \big\| \mathcal{W} - \hat{\mathcal{W}} \big\|_2 + (1 - \alpha) \big| \hat{\mathcal{W}} - \mathcal{W}' \big\|_2 \tag{7}$$

Here, the weight parameter $\alpha$ determines the contribution of each autoencoder's reconstruction error to the overall anomaly score. These reconstruction errors quantify the discrepancy between the input data and the corresponding reconstructed output of each autoencoder.

By appropriately combining the reconstruction errors from both $AE_1$ and $AE_2$, the anomaly score provides a comprehensive evaluation of anomalies at each time point. This approach capitalizes on the complementary capabilities of the two autoencoders, leading to improved accuracy and robustness in detecting anomalies within the system.

### 3.3. The Algorithm of Network Anomaly Detection

In this section, we provide a separate elucidation for both the training and inference processes. It is worth noting that the training process is divided into two distinct stages. The specifics of these training and inference operations are detailed as follows.

### 3.3.1. The Two-Phase Training Process

The specific training process is described in the following. The reconstruction errors for the two training phases can be specified as follows.

- **Phase 1: Autoencoder training.** Initially, the input data, represented by $W$, is subject to compression by the encoder, symbolized as $E$. This procedure reduces the input data into a latent space, denoted by $Z$. Following this, each of the two decoders embarks on the task of reconstruction, with the outcomes manifesting as $AE_1(W)$ and $AE_2(W)$, correspondingly. The errors incurred in this reconstruction process, can hence be individually articulated as follows:

$$Loss_{AE_1} = \| \mathcal{W} - AE_1(\mathcal{W}) \|_2 \tag{8}$$

$$Loss_{AE_2} = \| \mathcal{W} - AE_2(\mathcal{W}) \|_2 \tag{9}$$

- **Phase 2: Adversarial training.** Subsequent to the initial reconstruction by the two decoders, the output derived from $AE_1$ is subjected to another compression by $E$, transitioning once more into the latent space $Z$. This compressed output is then conveyed to $AE_2$ for another round of reconstruction, resulting in the output $AE_2(AE_1(W))$. This procedure allows $AE_1$ to be conditioned in such a way as to deceive $AE_2$, whereas $AE_2$ is trained to differentiate between the original data and the data generated from $AE_1$. Consequently, $AE_1$ aspires to minimize the disparity between the original data $W$ and the outputs from $AE_2$, while $AE_2$ endeavors to maximize this distinction. Within this adversarial framework, the reconstruction errors can be represented as follows:

$$Loss_{AE_1} = + \| \mathcal{W} - AE_2(AE_1(\mathcal{W})) \|_2 \tag{10}$$

$$Loss_{AE_2} = - \| \mathcal{W} - AE_2(AE_1(\mathcal{W})) \|_2 \tag{11}$$

Together, these two stages form a dynamic and robust training process that allows us to effectively compress and reconstruct the data while leveraging the adversarial mechanism to refine the quality of this reconstruction.

Summarily, both autoencoders serve dual roles. In the first phase, $AE_1$ aims to minimize the reconstruction discrepancy of the original data $W$, while in the second phase, it seeks to minimize the difference between $W$ and the reconstructed output derived from

$AE_2$. Similarly, $AE_2$ also endeavors to minimize the reconstruction error of $W$ during the first phase; however, in the second phase, it strives to maximize the reconstruction error of the input data as reconstructed by $AE_1$. As the training advances through multiple iterations, the reconstruction error per epoch can be articulated as follows:

$$Loss_{AE_1} = \frac{1}{n}\|\mathcal{W} - AE_1(\mathcal{W})\|_2 + \left(1 - \frac{1}{n}\right)\|\mathcal{W} - AE_2(AE_1(\mathcal{W}))\|_2 \tag{12}$$

$$Loss_{AE_2} = \frac{1}{n}\|\mathcal{W} - AE_2(\mathcal{W})\|_2 - \left(1 - \frac{1}{n}\right)\|\mathcal{W} - AE_2(AE_1(\mathcal{W}))\|_2 \tag{13}$$

where $n$ denotes the training epoch.

### 3.3.2. The Inference Process

During the inference phase, the input data are represented by an unknown window data, denoted by $\widehat{W}$. We propose to consider the reconstruction error obtained during this process as the anomaly score. This score can be mathematically expressed as:

$$S(\tilde{\mathcal{W}}) = \alpha\big|\tilde{\mathcal{W}} - AE_1(\tilde{\mathcal{W}})\big|_2 + (1 - \alpha)\big|\tilde{\mathcal{W}} - AE_2\big(AE_1(\tilde{\mathcal{W}})\big)\big|_2 \tag{14}$$

The specific values of $\alpha$ greatly influence the quantity of results that we deem positive predictions. Setting a smaller $\alpha$ value tends to increase the count of instances classified as positive. This stringent criterion is particularly suitable for scenarios requiring high detection sensitivity. Conversely, if a lower sensitivity is appropriate for the situation, we can decrease the instances predicted as positive by tuning these parameters.

These adjustable parameters allow the anomaly detection algorithm to be well suited for various circumstances, notably those of a 5G private network quality monitoring system, which often requires scenario-based adjustments. By effectively manipulating $\alpha$, the algorithm can better meet the distinctive needs of various anomaly detection applications within this context. For the ease of understanding, we summarize the anomaly detection algorithm in Algorithm 1.

---

**Algorithm 1** Anomaly detection algorithm.

---

**Input:** Historical window data $\mathcal{W} = \{\mathbf{w}_t\}_{t=1}^{T}$, new window data $\tilde{\mathcal{W}} = \{\tilde{\mathbf{w}}_t\}_{t=1}^{\tilde{T}}$, epoch $N$, threshold $\eta$, weight parameter $\alpha$

**Output:** Anomaly labels $\mathcal{L} = \{l_t\}_{t=1}^{\tilde{T}}$

1: Initialize $E$, $D_1$ and $D_2$
2: **for** $n = 1 \rightarrow N$ **do**
3:    **for** $t = 1 \rightarrow T$ **do**
4:        $Loss_{AE_1} = \frac{1}{n}\|\mathbf{w}_t - AE_1(\mathbf{w}_t)\|_2 + \left(1 - \frac{1}{n}\right)\|\mathbf{w}_t - AE_2(AE_1(\mathbf{w}_t))\|_2$
5:        $Loss_{AE_2} = \frac{1}{n}\|\mathbf{w}_t - AE_2(\mathbf{w}_t)\|_2 - \left(1 - \frac{1}{n}\right)\|\mathbf{w}_t - AE_2(AE_1(\mathbf{w}_t))\|_2$
6:        Update the weights of $E$, $D_1$ and $D_2$ using $Loss_{AE_1}$ and $Loss_{AE_2}$
7:    **end for**
8: **end for**
9: **for** $t = 1 \rightarrow \tilde{T}$ **do**
10:    $S(\tilde{\mathbf{w}}_t) = \alpha\|\tilde{\mathbf{w}}_t - AE_1(\tilde{\mathbf{w}}_t)\|_2 + (1 - \alpha)\|\tilde{\mathbf{w}}_t - AE_2(AE_1(\tilde{\mathbf{w}}_t))\|_2$
11:    **if** $S(\tilde{\mathbf{w}}_t) > \eta$ **then**
12:        $l_t = 1$
13:    **else**
14:        $l_t = 0$
15:    **end if**
16: **end for**
17: **return** $\mathcal{L} = \{l_t\}_{t=1}^{\tilde{T}}$

---

## 4. Network Anomaly Forecasting

In order to enable effective anomaly forecasting, it is necessary to predict the future network data based on historical performance data. The 5G vertical applications provide a diverse set of indicators, which can be used to conduct a comprehensive evaluation of network quality from multiple dimensions. These indicator data are presented in the form of continuous time series data, which can be utilized to predict the future network status and further achieve anomaly forecasting. To accomplish this, we employ a classic predictive network model called long short-term memory (LSTM) to predict the future network indicator data. LSTM is a type of recurrent neural network that is well suited for processing sequential data such as time series data. By training the LSTM model on historical performance data, we are able to generate accurate predictions of future network performance. In this section, the structure and algorithm of the network anomaly forecasting scheme are described in detail.

### 4.1. The Structure of Network Anomaly Forecasting

As shown in Figure 2, by leveraging the unique properties of LSTM as the prediction model, we can accurately predict future network indicators, which is critical for achieving the function of network anomaly forecasting.

#### 4.1.1. Network Data Prediction

In pursuit of forecasting anomalies in unlabeled network data, we tailored our approach to primarily focus on two crucial aspects. These elements reflect our strategic utilization of available historical data and the application of advanced predictive network modeling techniques. As we dive into these sections, we will be shedding light on the in-depth processes and methodologies used in our study.

The historical data of 5G vertical applications with length $T$ is utilized as input, where each certain time $t$ contains $M$ kinds of features. These input data provide us with a rich history of network performance that can be used to forecast future network activity. We predict future time series data of length $T_0$ from historical data, in which $T_0 < T$. This prediction allows us to project future network performance based on past performance and current network conditions. Once we have obtained the predicted data, we divide it into multiple time windows. This allows us to examine the network performance in discrete, manageable chunks, rather than as a continuous stream of data. Each time window represents a unique snapshot of network performance.

In the context of predictive network modeling, LSTM is a type of recurrent neural network (RNN) that has shown excellent performance in sequence prediction tasks due to its ability to retain information over a longer period of time. LSTM is composed of multiple memory blocks, each of which has a cell state and a hidden state that are used to generate the output related to the input.

At each time step $t$, the matrix of historical indicator data feature $X_t$ is fed into the LSTM network. The network then generates the cell state $C^t$ and hidden state $h^t$, which provide the LSTM network with a sort of "memory" of past network conditions. The previous cell state $C^{t-1}$ and hidden state $h^{t-1}$ are also utilized to compute $h^t$, allowing the network to retain information over time. This temporal memory is one of the key features that make LSTM networks so effective for time series prediction tasks.

In addition, LSTM memory blocks iteratively update themselves through the gating signals from the three gate controllers of the forget gate, input gate, and output gate. Each of these gates serves a unique function in the LSTM network: the forget gate determines which information to discard from the cell state, the input gate determines which information to update in the cell state, and the output gate determines which information to output as the final prediction.

Once we have obtained the predicted data, we conduct anomaly detection using the method introduced in Section 3 to achieve network anomaly forecasting.

4.1.2. Anomaly Forecasting Function

Utilizing the method introduced in Section 3, we conduct anomaly detection on these predicted network data. This involves comparing the predicted network performance with expected performance levels. Any deviations from the expected performance that exceed a certain threshold are considered anomalies.

Similarly, by calculating an anomaly score for each predicted time window, we are able to evaluate the quality of the future network and detect any potential anomalies. This anomaly score serves as a quantifiable measure of network performance, allowing us to objectively evaluate the health of the network.

By utilizing the anomaly score of the future network, we are able to achieve the function of anomaly warning. This proactive warning system alerts network administrators to potential problems before they have a significant impact on network performance. Specifically, if the anomaly score exceeds a certain threshold, the system can issue an alert to network administrators. These alerts indicate that corrective action may be necessary to prevent network downtime or degradation. This enables administrators to take proactive steps to address potential issues before they become more serious problems.

### 4.2. The Algorithm of Network Anomaly Forecasting

Utilizing the historical data, the data prediction model updates the cell state and hidden state, and forecasts the future network data. The cell state at the current moment determines which features are retained for transmission to the next memory block for prediction based on the input and hidden state at the current moment, while the hidden state decides which features of input data are not required for prediction and can be discarded. Obtaining the future network data as the input of anomaly detection model introduced in Section 3, we obtain the anomaly scores of future networks and achieve the function of anomaly warning. The anomaly warning algorithm is summarized in Algorithm 2 and the specific procedure of anomaly forecasting is as follows.

---

**Algorithm 2** Anomaly forecasting algorithm.

---

**Input:** Historical window data $\mathcal{W} = \{\mathbf{w}_t\}_{t=1}^{T}$, threshold $\eta$, weight parameter $\alpha$
**Output:** anomaly labels of predicted data $\left\{\hat{l_1}, \ldots, \hat{l_{T_0}}\right\}$

1: Initialize $C_0$ and $h_0$
2: **for** $t = 1 \rightarrow T$ **do**
3:      $Z_f = f_{sig}(w_f * f_{jt}(X^t, h^{t-1}))$
4:      $Z = f_{tanh}(w * f_{jt}(X^t, h^{t-1}))$
5:      $Z_i = f_{sig}(w_i * f_{jt}(X^t, h^{t-1}))$
6:      $C^t \leftarrow Z_f \odot C^{t-1} + Z_i \odot Z$
7:      $Z_o = f_{sig}(w_o * f_{jt}(X^t, h^{t-1}))$
8:      $h_t \leftarrow Z_o \odot f_{tanh}(C^t)$
9:      $\hat{Y}_t = f_{sig}(w' * h^t)$
10: **end for**
11: **for** $t = 1 \rightarrow T_0$ **do**
12:      $S(\hat{Y}_t) = \alpha \left\| \hat{Y}_t - AE_1(\hat{Y}_t) \right\|_2 + (1 - \alpha) \left\| \hat{Y}_t - AE_2(AE_1(\hat{Y}_t)) \right\|_2$
13:      **if** $S(\hat{Y}_t) > \eta$ **then**
14:          $\hat{y}_t = 1$
15:      **else**
16:          $\hat{y}_t = 0$
17:      **end if**
18: **end for**
19: **return** $\left\{\hat{l_1}, \ldots, \hat{l_{T_0}}\right\}$

---

- **Data Input**: Utilizing the historical data in the past time as the input, the input gate of a memory block in LSTM can make full use of temporal information in all time points

to identify changes in features and provide a basis for subsequent feature screening by updating the cell state. In other words, input gates $Z_i$ determine what feature from the network input $X_t$ at the current moment is saved to the cell state $C_t$. How it updates is shown as follows:

$$Z = f_{tanh}(w * f_{jt}(X^t, h^{t-1})) \tag{15}$$

$$Z_i = f_{sig}(w_i * f_{jt}(X^t, h^{t-1})) \tag{16}$$

$$C^t = Z_f \odot C^{t-1} + Z_i \odot Z \tag{17}$$

where Z represents the current information of input and $\odot$ represents the Hadamard product. $f_{tanh}$ represents the activation function called the hyperbolic tangent function, while $w$ and $w_i$ are the weight matrix for current information $Z$ and input information $Z_i$

- **Feature screening**: In the time windows of input, there are some KPI features that contribute little to prediction because they are correlated to other features, which can be inferred by other important features. Thus, a forgetting gate can filter out the features that contribute more to prediction, where the weight matrix $w_f$ is used to achieve this function, and then combine these features and input information at the current moment to complete the prediction. The forgetting gate $Z_f$ determines how much of the unit state $C_{t-1}$ at the previous moment is retained to the current state $C_t$. The definition of a forgetting gate is:

$$Z_f = f_{sig}(w_f * f_{jt}(X^t, h^{t-1})) \tag{18}$$

where $w_f$ is the weight matrix for forgetting information $Z_f$, and $f_{sig}$ means that $X^t$ and $h^{t-1}$ are concatenated together in a column and $f_{sig}$ is the activation function sigmoid.

- **Predicted results output**: Combining the filtered features and current input, the memory block output the predicted data at current time and hidden state through output gate. The hidden state is transmitted to the next memory block to predict the data of the next time point. After traversing multiple memory blocks, the final memory block receives the cell state and hidden states which contain all the information on historical time, and it outputs the final result of prediction. The output gate $Z_o$ decides the next $h_t$, which will be pass with the new cell state $C^t$ to the next memory block. The process of updating is presented as follows:

$$Z_o = f_{sig}(w_o * f_{jt}(X^t, h^{t-1})) \tag{19}$$

$$h_t = Z_o \odot f_{tanh}(C^t) \tag{20}$$

$$Y_t = f_{sig}(w' * h^t) \tag{21}$$

where $Y^t$ is the output related to $X^t$ and $w'$ is the weight matrix for $Y_t$.

## 5. Experiments and Results

### 5.1. Experimental Setup

In this study, we examine the performance of three distinct 5G vertical applications, namely the Internet of Vehicles, Industrial Internet, and intelligent manufacture. The data are collected from the practical operation of the experimental 5G network for the three aforementioned typical vertical applications, which are the most important scenarios of 5G vertical applications for China Unicom that can provide enough network data for generating high-quality deep learning models. Moreover, unlike the conventional

evaluation methods by generating data via software simulation, our experiment results are more convincing. Finally, our three scenarios studied herein have diverse requirements in terms of user experiences and network quality, and thus our experiment results can cover most of typical 5G vertical applications, which can provide more useful insights for the deployment potential of our proposed scheme. To this end, we obtained a total of 10,500 data samples from each of the three scenarios for seven days, of which the first 80% were used as the training dataset and the remaining data were used as the test dataset, then utilized to evaluate the quality detection and forecasting performance. We select several metrics with the highest correlation from the derived multi-metric data, and process the multi-dimensional time series data to serve as input data for quality monitoring. Specifically, concerning the 5G key performance indicators (KPIs), we focus on the airport traffic, wireless resource utilization, average user uplink/downlink rates, CQI, RRC connection success rate, and the packet loss rate.

For the parameters in the anomaly detection algorithm, the window data length is set at 12 and the number of epochs is set at 100. Additionally, the parameters $\alpha = 0.5$ is also utilized. When performing tests, we search for the optimal threshold $\lambda$ by utilizing the ROC curve. We note that the optimal value is crucial in terms of enhancing the overall performance of the model. The LSTM model is a three-layer neural network. The first two layers are with 64 and 32 neurons, respectively, and a fully connected layer is employed as the output layer. The ReLu function is used as the activation function. During the training process, the size of the mini-batch and learning rate are set to 32 and 0.001, respectively.

To assess the efficacy and generalization of our proposed approach, we conducted an anomaly detection experiment on network data in three distinct settings. The performance evaluation of our method was based on three well established metrics: precision (*P*), recall (*R*), and F1-score (*F*1). Among these metrics, precision denotes the fraction of actual anomaly samples identified as such by our method, while recall indicates the fraction of true anomaly samples correctly classified as "abnormal" by our algorithm. To provide a comprehensive picture of the detection performance, we further calculated the F1-score, which balances precision and recall. The use of these evaluation metrics enabled us to quantitatively gauge the effectiveness and robustness of our method across different scenarios. They can be represented as

$$P = \frac{TP}{TP + FP}, \quad R = \frac{TP}{TP + FN}, \quad F1 = 2 \cdot \frac{P \cdot R}{P + R} \tag{22}$$

where *TP* denotes true positives; *FP* denotes false positives; and *FN* denotes false negatives.

### 5.2. 5G Vertical Applications Platform

The specific details about the 5G vertical applications platform, such as the exact source of their network data (data access interfaces) and the detailed characteristics of various application scenarios (e.g., the number of base stations per scenario), vary depending on the specific implementation, standards, and configurations defined by different operators and technology providers. Generally speaking, network data from 5G dedicated networks can come from several sources, including:

- **Network equipment:** This includes data collected from base stations, switches, routers, and other hardware elements that make up the network infrastructure.
- **Network management systems:** These systems monitor and manage the operation of the network, and can provide data about network traffic, faults, performance, and security.
- **User equipment:** This includes data from user devices connected to the network, which can provide data on usage patterns, performance, and service quality.

As for the specific scenarios and the number of base stations involved, this would greatly depend on the needs of the application being supported by the dedicated network. For example, a dedicated network supporting an industrial automation application in a factory might be quite dense with a large number of base stations to provide high-capacity,

low-latency communication within a small area. Conversely, a dedicated network for a rural telecommunication service might have a much lower density of base stations covering a larger geographic area.

To obtain the most accurate and up-to-date information, one may refer to the technical specifications and guidelines provided by the relevant standards bodies (like 3GPP for 5G), technology providers, and network operators, or consulting with experts in the field.

### 5.3. Performance of the Proposed Anomaly Detection Method

The results depicted in Figures 3–5, which were obtained by simulating the proposed network anomaly detection model with a weight parameter value of $\alpha = 0.5$, demonstrate the convergence behavior of the algorithm. It can be observed that, over time, the algorithm gradually converges towards a stable state. The convergence process is characterized by a decreasing trend in the loss function, as shown in Figures 3–5. In intelligent manufacture, convergence is slightly less effective due to a slight lack of data volume. Remarkably, the algorithm exhibits an impressive convergence speed, achieving convergence in less than 10 epochs for intelligent manufacturing and Industrial Internet. This rapid convergence is a notable advantage of the proposed network anomaly detection model. By converging quickly, the algorithm is able to efficiently and effectively detect and identify network anomalies. This attribute is particularly valuable in real-time network monitoring scenarios where prompt anomaly detection is crucial for maintaining network security and stability.

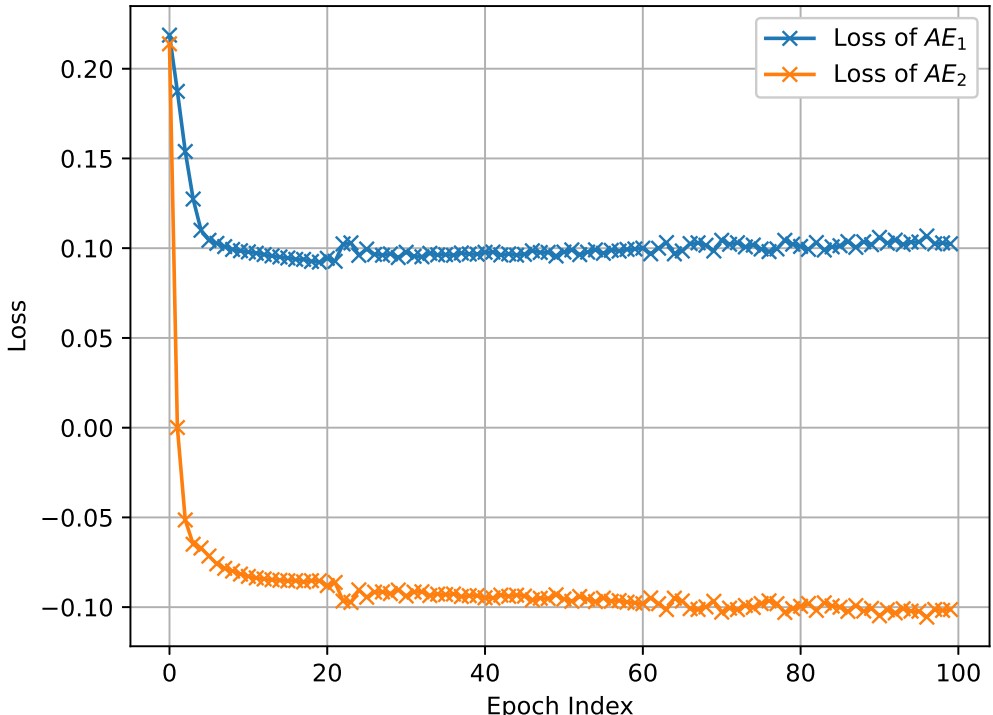

**Figure 3.** The loss of network anomaly detection algorithm in the Internet of Vehicles.

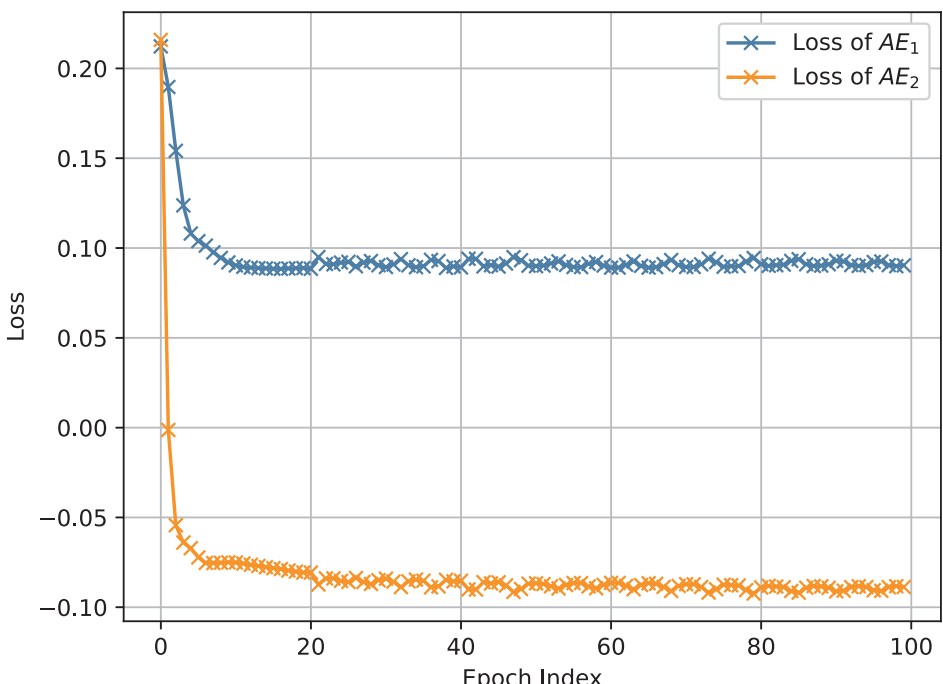

**Figure 4.** The loss of network anomaly detection algorithm in intelligent manufacturing.

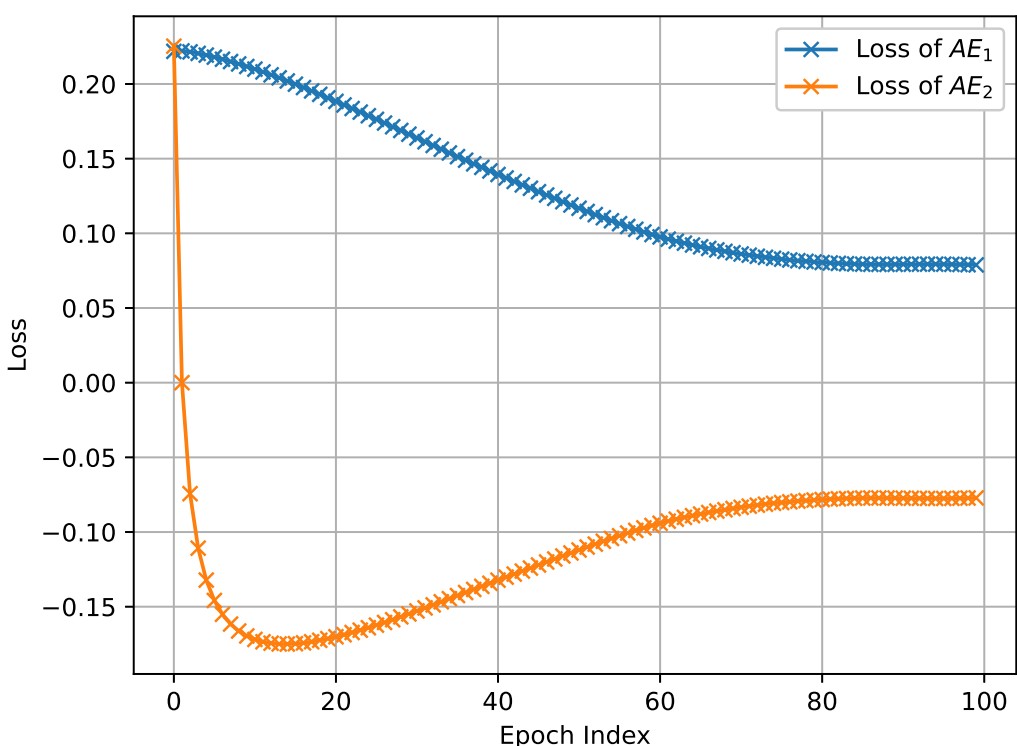

**Figure 5.** The loss of network anomaly detection algorithm in the Industrial Internet.

Furthermore, we scrutinize the performance of anomaly detection based on precision, recall, and F1-score using network data derived from three distinct 5G vertical applications. A critical factor in this analysis is the role of the weight parameter in the calculation of the anomaly score. To fully understand its impact, we initially fine-tune this parameter within our proposed anomaly detection method, taking into consideration the distribution of network data across different application scenarios.

Table 2 provides a comprehensive summary of the accuracy of anomaly detection in network data across different small regions within three distinct application scenarios: Internet of Vehicles, intelligent manufacturing, and Industrial Internet. The table presents the results obtained using various weight parameter settings.

**Table 2.** The performance of the proposed network anomaly detection method for three 5G vertical applications: Internet of Vehicles, intelligent manufacturing and Industrial Internet.

| Cell Index | The Proposed Scheme | | | | Autoencoder-Only | | | |
| --- | --- | --- | --- | --- | --- | --- | --- | --- |
| | P | R | F1 | Average F1 | P | R | F1 | Average F1 |
| Internet of Vehicles | | | | | | | | |
| 0 | 0.933 | 0.921 | 0.927 | | 0.836 | 0.821 | 0.828 | |
| 1 | 0.938 | 0.913 | 0.925 | | 0.827 | 0.831 | 0.829 | |
| 2 | 0.917 | 0.941 | 0.929 | 0.926 | 0.851 | 0.817 | 0.834 | 0.830 |
| 3 | 0.926 | 0.926 | 0.926 | | 0.826 | 0.832 | 0.829 | |
| 4 | 0.942 | 0.908 | 0.925 | | 0.831 | 0.826 | 0.828 | |
| Intelligent manufacturing | | | | | | | | |
| 0 | 0.864 | 0.919 | 0.891 | | 0.811 | 0.876 | 0.842 | |
| 1 | 0.863 | 0.923 | 0.892 | | 0.817 | 0.874 | 0.845 | |
| 2 | 0.860 | 0.937 | 0.897 | 0.894 | 0.815 | 0.871 | 0.842 | 0.840 |
| 3 | 0.856 | 0.939 | 0.896 | | 0.809 | 0.867 | 0.837 | |
| 4 | 0.849 | 0.944 | 0.894 | | 0.808 | 0.866 | 0.836 | |
| Industrial Internet | | | | | | | | |
| 0 | 0.861 | 0.918 | 0.889 | | 0.796 | 0.867 | 0.830 | |
| 1 | 0.858 | 0.925 | 0.890 | | 0.795 | 0.865 | 0.829 | |
| 2 | 0.853 | 0.927 | 0.888 | 0.888 | 0.791 | 0.862 | 0.825 | 0.825 |
| 3 | 0.849 | 0.928 | 0.887 | | 0.786 | 0.859 | 0.821 | |
| 4 | 0.847 | 0.932 | 0.887 | | 0.785 | 0.857 | 0.819 | |

The findings indicate that anomaly detection using network data from these three application scenarios consistently achieves a high F1-score. This suggests that the proposed anomaly detection algorithms effectively identify and classify anomalies in diverse network environments. Moreover, the average F1-score across all scenarios indicates a generally high level of accuracy in the detection of network anomalies.

Furthermore, the consistent performance across the three application scenarios suggests that the proposed approach is robust and adaptable. The ability to achieve accurate anomaly detection in various IoT contexts is vital for maintaining network security and integrity. These results contribute to the advancement of anomaly detection techniques and provide valuable insights for network administrators and researchers in the field of IoT security.

In order to provide a more comprehensive evaluation of the proposed anomaly detection method, we compare its performance against a baseline model, i.e., autoencoder-only method. This baseline model serve as a standard measure to evaluate the accuracy of detection using the F1-score as the metric. For the purposes of this comparison, we set the weight parameter $\alpha$ used in the calculation of the anomaly score in our proposed method to be 0.5.

Table 2 visually presents the performance of anomaly detection across three distinct application scenarios: Internet of Vehicles, intelligent manufacturing, and Industrial Internet. Here, our proposed anomaly detection approach is contrasted with the autoencoder-only and adversarial-only methods. The data displayed in Table 2 lead us to conclude that, irrespective of the application scenario, our proposed network anomaly detection method consistently outperforms the autoencoder-only method in terms of detection accuracy. This superior performance can be ascribed to our method's integration of the autoencoder and

the GAN. By leveraging the strengths of both networks, our proposed method enhances detection precision, surmounting limitations such as GAN non-convergence.

These results provide strong support for the effectiveness of our proposed anomaly detection method. By bridging the capabilities of the autoencoder and GAN, the proposed method not only improves the detection accuracy across various 5G vertical applications, but also mitigates some of the key challenges commonly faced in network anomaly detection. However, as with all models, the performance of the proposed method could be influenced by various factors, and thus further investigations are warranted to fully understand its potential and limitations in different contexts.

### 5.4. Performance of the Proposed Anomaly Forecasting Method

In this subsection, we evaluate the performance of the proposed anomaly forecasting method in terms of accuracy, using the F1-score as the evaluation metric. To examine the advantages of the proposed anomaly forecasting method, we apply it to forecast anomalies in network data from three different 5G vertical applications. Table 3 presents the test accuracy for datasets obtained from the Internet of Vehicles, intelligent manufacturing, and the Industrial Internet. We compare the proposed anomaly forecasting method with a baseline approach using historical network data as network data at future periods for anomaly detection. It can be observed that the accuracy of the predicted results is comparable to the baseline in most cases, and in some instances, it may even surpass the baseline approach.

**Table 3.** The performance of the proposed network anomaly forecasting method for three 5G vertical application.

| Method | F1 | |
|---|---|---|
| | The Proposed scheme | The Baseline Scheme |
| Internet of Vehicles | 0.912 | 0.701 |
| Intelligent manufacturing | 0.899 | 0.687 |
| Industrial Internet | 0.869 | 0.669 |

In this subsection, Table 3 elucidates the test accuracy for datasets hailing from the Internet of Vehicles, intelligent manufacturing, and Industrial Internet verticals. In our study, we juxtapose the proposed anomaly forecasting method with a baseline approach that directly harnesses real data for anomaly detection.

Upon the close inspection of Table 3, we can observe that, in most instances, the accuracy of the forecasted results closely aligns with, if not outperforms, the baseline approach. This underscores the effectiveness of the proposed anomaly forecasting method, as it showcases an ability to yield accuracy levels that are not just on par with, but occasionally surpass, the performance of the baseline approach.

These results are quite encouraging, suggesting that the proposed anomaly forecasting method holds promise in enhancing anomaly detection in 5G vertical applications. The method's ability to consistently match or outdo the accuracy of the baseline approach underlines its potential as a valuable tool for 5G network data anomaly detection and forecasting. However, it is worth noting that the effectiveness of the anomaly forecasting method could vary depending on the specific characteristics of the dataset and the nature of the anomalies present, necessitating further investigations into its applicability across a wider range of scenarios.

In order to provide a more explicit evaluation of our proposed anomaly prediction method, we conducted a 24-hour-per-day case study aimed at highlighting the differences in the anomaly scores obtained based on real versus predicted data. Figures 6–8 depict the anomaly scores for anomaly detection using real or predicted network data from the Internet of Vehicles, intelligent manufacturing, and Industrial Internet, respectively.

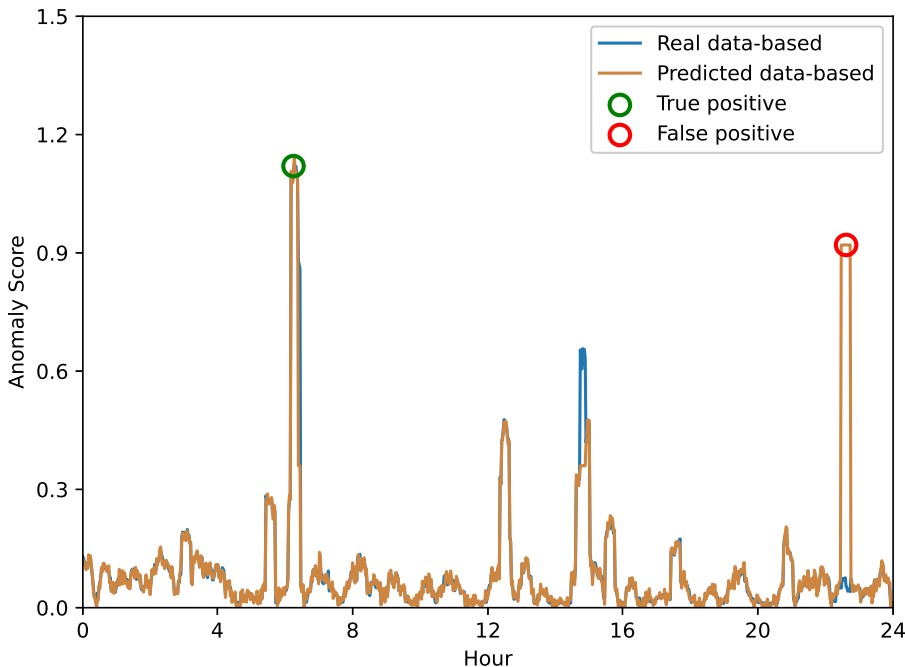

**Figure 6.** The anomaly score of network anomaly forecasting in the Internet of Vehicles.

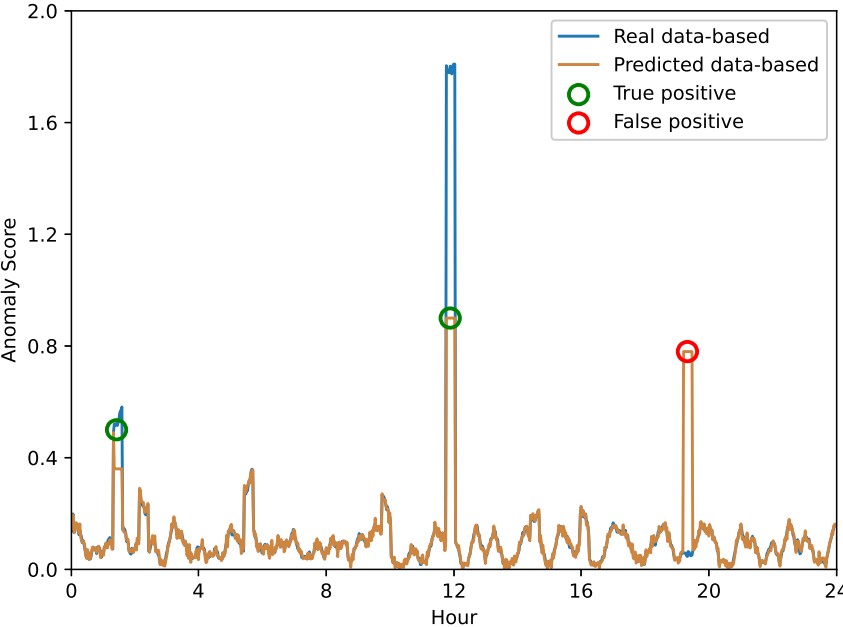

**Figure 7.** The anomaly score of network anomaly forecasting in intelligent manufacturing.

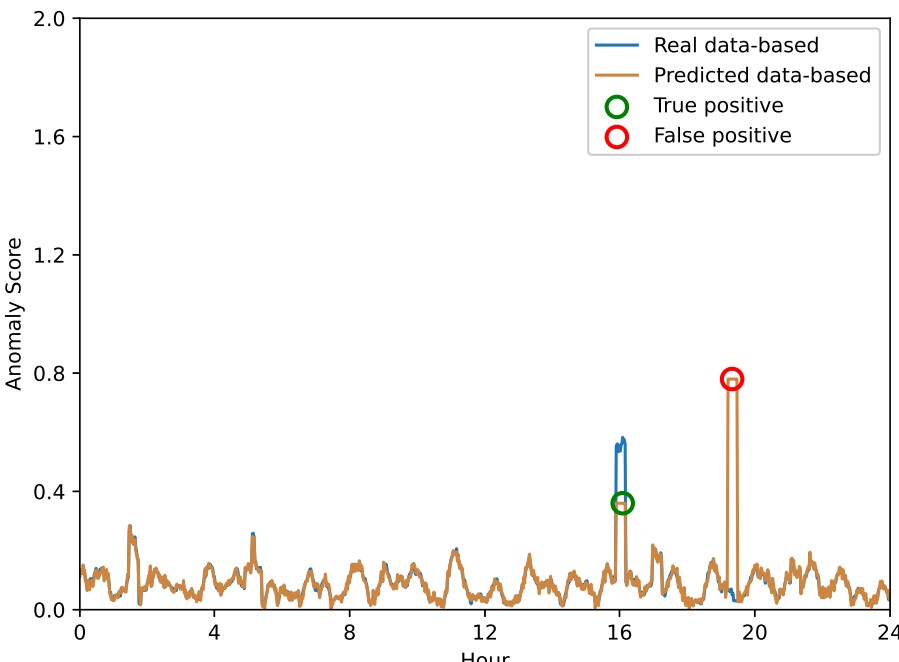

**Figure 8.** The anomaly score of network anomaly forecasting in the Industrial Internet.

A closer look at these results shows that the performance of anomaly prediction results is comparable to direct anomaly detection. However, it is important to note that anomaly warnings can be subject to false detections. As shown in Figure 6, the Internet of Vehicles shows that in some instances, the anomaly scores of both methods spike dramatically. These instances represent potential anomalies in the system, highlighting the ability of anomaly prediction and anomaly detection to flag these instances. In contrast, in other cases, we can observe that one method may flag potential anomalies while the other does not, indicating a difference in the detection capabilities of the two methods. Similarly, the anomaly scores for the intelligent manufacturing and Industrial Internet data shown in Figures 7 and 8 highlight the same interesting phenomenon. While the overall trends in anomaly scores for both methods are very close, there are still significant differences. These inconsistencies remind us of the inherent limitations of each method—some anomalies may be misidentified.

In summary, while our proposed anomaly forecasting method demonstrates a comparable performance in direct anomaly detection, it also highlights the unique advantages and shortcomings of each approach. These findings suggest the need for further research into optimizing and possibly integrating these two methodologies to maximize anomaly detection and forecasting accuracy.

## 6. Conclusions

In this paper, a framework for network quality monitoring has been proposed for both network anomaly detection as well as anomaly forecasting functions. The framework is designed to model the time dependence of unlabeled network data. An anomaly detection method is first used for anomaly detection, combining the autoencoder with GAN while compensating for the stability problems associated with GAN. Then, the LSTM model is used to predict network quality data and detect anomalies in the predicted data using the anomaly detection model to achieve anomaly forecasting. Finally, the proposed quality monitoring method is demonstrated through simulation experiments on network data measured in various 5G vertical scenarios, an our proposed method can provide 10%/5%/6% F1-score gain in terms of network anomaly detection and 21%/21%/20% F1-score gain in

terms of network anomaly forecasting in three scenarios, highlighting its superior accuracy and stability.

**Author Contributions:** Q.Z. designed the experiment scheme, analyzed the experimental results, and wrote the paper. T.Z., B.C., T.G. and K.C. made gait experiments. X.L., Y.D. and Z.Z. modified this paper. All authors have read and agreed to the published version of the manuscript.

**Funding:** This paper was support by National Natural Science Foundation (Grant No. 61971061), Beijing Natural Science Foundation (Grant No. L223026), and 5G Evolution Wireless Air interface Intelligent R&D and Verification Public Platform Project (Grant No. 2022-229-220).

**Institutional Review Board Statement:** Not applicable.

**Informed Consent Statement:** Not applicable.

**Data Availability Statement:** Not applicable.

**Acknowledgments:** The authors would like to express their sincere appreciation to the editors and reviewers for their invaluable assistance in bringing this work to fruition.

**Conflicts of Interest:** The authors declare no conflicts of interest and the funders had no role in the design of the study; in the collection, analyses, or interpretation of data; in the writing of the manuscript; or in the decision to publish the results.

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
