# Peer review of "Generative Adversarial Network-Based Anomaly Detection and Forecasting with Unlabeled Data for 5G Vertical Applications"

_applsci, doi:10.3390/app131910745_

Round 1

Reviewer 1 Report

The article "Generative Adversarial Network-Based Anomaly Detection and Forecasting with Unlabeled Data for 5G Vertical Applications" present an implementation of network quality anomaly detection in different 5G scenarios using a fast and stable anomaly detection model. In this reviewer’s opinion, the article needs to be improved:

1) Explain how the proposed work can address issues in South America (10.1007/978-3-031-04435-9_21) and other regions with problems expanding the 5G network.

2) Correct the Table 2 title.

3) Increase the size of Figure 3.

4) Insert the hyperparameters used in LSTM

Author Response

Please refer to our uploaded PDF version response letter.

Reviewer 2 Report

The topic is relatively new and will increase in importance with the passage of time and the spread of technology.
There are many mathematical equations and symbols, and it is preferable to refer to them in one list or table.
The method of displaying the article can be improved by entering more visualization figures, especially in the first part of the research, in order to facilitate the delivery of the general idea before going into details.
Isn't there any reference in the research to the past year, 2022, or even 2023?
Abstract writing can be improved.
All notes are on the manuscript directly as notes.

The quality of writing in English is good and can be improved by conveying the general idea of the paragraphs by adding some effects, such as visualization figures.

Author Response

(The authors gave the same response as above.)

Reviewer 3 Report

The discussion about "Anomaly Detection and Forecasting with Unlabeled Data for 5G Vertical Applications" contribute knowledge especially application on 5G network.

1. Specific method should mention clearly in the abstract, instead of say good method.

2. Figure 3, 4, 5 and 6 should to enlarge for clearly view and readable graph.

3. The anomaly prediction as mention in section 4.2 that used LSTM algorithm with refer to algorithm 2, how many data simulated and traning and testing data, then how the acceptance percentage of results have to mention and discuss.

4. The case have done in simulation only 3 application in anomaly forecasting, which are internet of vehicles, industrial internet and intelligent manufacture.

5. Figure 5 and 6 typo "manufacture" and figure 5 should industrial internet ?

6. Conclusion need to add some results information, percentage, performance and etc.

Author Response

(The authors gave the same response as above.)

Round 2

Reviewer 1 Report

Nothing to add